# Discriminant Analysis and Data Mining CHAID Decision Tree as Tools to Evaluate the Buffering Effect of Hydroxytyrosol on Reactive Oxygen Species in Rooster Sperm Cryopreservation

**DOI:** 10.3390/ani13193079

**Published:** 2023-10-02

**Authors:** Esther Díaz Ruiz, Antonio González Ariza, José Manuel León Jurado, Ander Arando Arbulu, Alejandra Bermúdez Oria, África Fernández Prior, Juan Vicente Delgado Bermejo, Francisco Javier Navas González

**Affiliations:** 1Department of Genetics, Faculty of Veterinary Medicine, University of Cordoba, 14014 Cordoba, Spain; estherddrr@gmail.com (E.D.R.); anderarando@hotmail.com (A.A.A.); juanviagr218@gmail.com (J.V.D.B.); fjng87@hotmail.com (F.J.N.G.); 2Agropecuary Provincial Centre, Diputación Provincial de Córdoba, 14071 Cordoba, Spain; jmlj01@dipucordoba.es; 3Instituto de la Grasa, Consejo Superior de Investigaciones Científicas (CSIC), 41013 Sevilla, Spain; aleberori@ig.csic.es (A.B.O.); mafprior@ig.csic.es (Á.F.P.)

**Keywords:** antioxidant, cryopreservation, reactive oxygen species, native avian breed, semen

## Abstract

**Simple Summary:**

For the conservation of genetic resources in avian species, semen freezing is very helpful. The disadvantage of this process is that spermatozoa suffer different types of damage. Exogenous antioxidants can be added to the cryopreservation extender to mitigate this damage. This study aimed to test whether the addition of different hydroxytyrosol (HT; an antioxidant derived from olive oil) concentrations produces beneficial effects in the sperm of a local avian breed (Utrerana roosters). For this purpose, different semen quality parameters, which fell under the following macro-areas were evaluated in both fresh and thawed semen: motility, morphology, membrane functionality, and flow-cytometry-related traits. At the statistical level, a descriptive analysis and a canonical discriminant analysis were performed, which allowed us to extract valuable information about the different studied variables. Lastly, a chi-squared automatic interaction detection (CHAID) decision tree (DT) was carried out and the reactive oxygen species (ROS) variable was found to have the highest power to discriminate between the different treatments according to the HT concentration. Low or no HT concentrations resulted in higher ROS values, and therefore, possible mechanical damage unrelated to plasma membrane peroxidation can be produced in the frozen–thawed rooster spermatozoa.

**Abstract:**

Sperm cryopreservation is effective in safeguarding genetic biodiversity in avian species. However, during this process, spermatozoa are very susceptible to plasma membrane peroxidation in the presence of high concentrations of reactive oxygen species (ROS). To mitigate this effect, the addition of exogenous antioxidants, such as hydroxytyrosol (3,4-dihydroxyphenylethanol; HT), an antioxidant derived from olive oil, to the cryopreservation sperm diluent, could be useful. To verify this, a cryopreservation diluent was supplemented with different concentrations (0 μg/mL, 50 μg/mL, 100 μg/mL, and 150 μg/mL) of HT. For this, semen was collected in 10 replicates from 16 roosters of the Utrerana avian breed, and a pool was prepared with the optimum quality ejaculates in each replicate. After cryopreservation, spermatozoa were thawed and different in vitro semen quality parameters were evaluated. A discriminant canonical analysis (DCA) was carried out and revealed that total motility (TM; Lambda = 0.301, F = 26,173), hypo-osmotic swelling test (HOST; Lambda = 0.338, F = 22,065), and amplitude of lateral head displacement (ALH, Lambda = 0.442; F = 14,180) were the variables with the highest discriminant power. Finally, a chi-squared automatic interaction detection (CHAID) decision tree (DT) was performed excluding fresh semen samples and ROS was found to be the most valuable variable to discriminate between the different established freezing groups. Samples in the absence of HT or with low concentrations of this antioxidant showed less desirable ROS values in cryopreserved rooster semen. The present study could lead to the improvement of cryopreservation techniques for the genetic material of local poultry breeds and optimize the conservation programs of endangered native avian breeds.

## 1. Introduction

Semen cryopreservation is an effective method to preserve the genetic resources of different aviary breeds [1]. This process is currently the only way for the ex situ in vitro storage of avian reproductive cells as the high yolk concentrations in avian eggs do not allow cryopreservation of oocytes [2,3]. In this way, one of the fundamental tools for the in vitro conservation of endangered breeds and high genetic value individuals is the development of germplasm banks, which are also a guarantee against the risk of epizootic diseases such as Avian Influenza [1].

However, the freezing–thawing process causes osmotic changes and the formation of intracellular ice crystals resulting from the imbalance between antioxidant and pro-oxidant defenses which, in turn, increase the proportion of reactive oxygen species (ROS) [4]. As a consequence, sperm membrane integrity and permeability are compromised and this process ultimately affects fertilization capacity and sperm viability [5]. The high vulnerability of rooster spermatozoa to membrane peroxidation is due to the large amounts of polyunsaturated fatty acids in the membrane compared to mammals, especially docosatetraenoic acid (22:4n - 6) and arachidonic acid (20:4n - 6) [6,7].

To counteract this effect, avian seminal plasma contains several antioxidant enzymes such as superoxide dismutase, catalase, and glutathione peroxidase, as well as other antioxidant compounds such as vitamins C and E, pyruvate, glutathione, and carnitine [8].

However, during cryopreservation, lipid peroxidation increases, which implies an attenuation of the aforementioned endogenous system, and the protection provided may be insufficient. In this regard, the addition of exogenous antioxidants is of interest, as there are a large number of antioxidant compounds that differ in their mechanisms of action, toxicity, and effectiveness [9,10].

Various plant extracts have antioxidant properties due to their high content of polyphenols, flavonoids, carotenes, gallic acid, tannins, and essential oils, which have major advantages over synthetic antioxidants as they generate fewer residues and cause less cytotoxicity [11].

Specifically, hydroxytyrosol (3,4-dyhydroxyphenylethanol, HT) is a phenolic compound derived from olive oil (*Olea europea*), which has antioxidant effects by reducing low-density lipoprotein oxidation, protecting against H_2_O_2_ cytotoxicity, and minimizing lactate dehydrogenase activity [12,13,14,15]. Specifically, this substance is obtained from alperujo olive pulp, which is a by-product generated in the two-phase system used in olive oil extraction [16]. Given that the sustainable production of olive oil is currently a matter of general interest, and even more so in the region of Andalusia (southern Spain) where this sector is very strong, this extraction system represents a step forward in the industrial phase since as through this system, the pollutant load of the wastewater is considerably reduced [17]. Therefore, the search for alternative uses for this by-product such as its use as an antioxidant would contribute to the optimization of this agricultural process [18].

The effect of an antioxidant can be measured across multiple semen quality variables. The most important parameters are sperm motility, viability, morphology, and concentration [19], and the use of statistical tools plays a pivotal role in determining which variables have the most discriminant power. Through discriminant canonical analysis (DCA), all the variables that are deemed appropriate can be considered simultaneously when distinguishing different populations [20]. DCA is a dimension-reduction technique that is derived from canonical correlation and principal component analysis [21]. As for principal component analysis (PCA), this statistical tool allows the development of a predictive model from a set of data showing the existing relationships between two or more variables [22]. However, while PCA aims to explain the maximum amount of variance, the discriminant functions are generated to maximize the difference between groups. Once discriminant functions have been computed, their structure could be examined to identify the different variables that contribute most to the discrimination between the semen-freezing treatments [23]. However, the complexity of the application of DCA produces considerable confusion among some researchers concerning its assumption, statistical properties, and data requirements [24].

Alternatively, the chi-squared automatic interaction detection (CHAID) decision tree (DT) method is one of the most used data mining models. This method, belonging to the category of non-parametric methods, allows classification, prediction, regression, estimation, description, visualization, and dimensionality reduction [25]. However, the use of CHAID DT presents certain methodological problems that should be solved. Model parameters should be specified before or during the application of the given algorithm. However, the main observed problems belong to different processes such as the rules for heterogenous populations splitting into smaller homogenous groups, the stopping criteria of the recursive process, and tree pruning [26].

Therefore, the present work aimed to determine the antioxidant effect of different HT concentrations on the post-thaw quality of Utrerana endangered avian breed semen using N-methylacetamide (NMA) as a cryoprotective agent. For this, DCA was used to determine the differential clustering patterns of different sperm-freezing treatments according to the antioxidant concentrations and quantify the contribution of semen quality-related traits to each cryopreservation treatment.

## 2. Materials and Methods

### 2.1. Ethical Approval

Protocols and animals used in the present research were managed following the prescriptions and regulations of the European Union (2010/63/EU) in its transposition to Spanish law (RD 53/2013). The data analyzed in this study were not obtained by experimental procedures but were obtained in the daily framework of an avian reproduction center and a center for the conservation of native breeds (both in the Agropecuary Provincial Center; Diputación de Córdoba). Thus, the present study is not covered by the legislation on the protection of animals used for scientific purposes and is outside the scope of evaluation of the Ethics Committee of the University of Córdoba.

### 2.2. Animals

The study was conducted at the Agropecuary Provincial Center of the Diputación of Córdoba (Spain). A total of 16 roosters of the Utrerana avian breed (1–3 years old) were used in the study. Roosters were housed in individual cages (95 cm × 95 cm × 95 cm) under a natural photoperiod.

All animals received a commercial diet (15.20% crude protein, 4.60% crude fat and oils, 3.20% crude fiber, 14.00% crude ashes, 4.10% calcium, 0.66% phosphorus, 0.19% sodium, 0.31% methionine, 0.72% lysine) and water *ad libitum*.

### 2.3. Semen Collection

From each rooster, one ejaculate was collected twice a week (from October to November 2021) through the abdominal massage technique described by Burrows and Quinn [27]. A total of 10 ejaculates per bird (160 ejaculates in total) were collected. All individual ejaculates of acceptable quality were pooled for each day of semen evaluation. The minimum seminal quality criteria were as follows: volume (>0.2 mL), concentration (>3 × 109 spz/mL), motility (≥80%), and morphology (≤10 to 15%). Semen was collected individually in 2 mL Eppendorf tubes for subsequent pooling of all ejaculates in 15 mL Falcon tubes if each of them passed the minimum inclusion criteria.

### 2.4. Sperm Dilution, Freezing, and Thawing

After pooling, the semen was cooled for one hour to 5 °C in a programmable cooler (Cell incubator SH-020S, Welson, Seoul, Republic of Korea) and the temperature was reduced at a rate of 0.3 °C/min.

To determine the effect of HT on the quality of thawed semen, semen samples were diluted when they reached 5 °C with a diluent containing different HT concentrations to obtain four different treatments according to the antioxidant concentration: control (diluted semen without antioxidant), HT1 (50 μg/mL), HT2 (100 μg/mL), and HT3 (150 μg/mL). The diluent was added in two steps. Thus, to obtain the final HT concentration, the required antioxidant amount was divided into two parts.

A first dilution was performed with a diluent (Fraction A) composed of 0.2 g D-(+)-glucose, 3.8 g D-(+)-trehalose dihydrate, 1.2 g L-glutamic acid monosodium salt, 0.3 g potassium acetate, 0.08 g magnesium acetate tetrahydrate, 0.05 g sodium citrate tribasic dihydrate, 0.4 g BES, 0.4 g Bis-Tris, and 0.001 g gentamicin sulfate (pH = 6.8, osmolarity = 360 mOsm) [28]. After 30 min, a second dilution was performed with the same diluent containing 18% NMA (final concentration 9%) as cryoprotectant (Fraction B).

Within 10 min of the second dilution, samples were packed in 0.25 mL straws with a final concentration of 500 × 10^6^ spz/straw and placed in nitrogen vapor at a height of 4 cm for 30 min. After the 30 min had elapsed, straws were immersed in liquid nitrogen (−196 °C) until their use. For thawing, the straws were immersed in a water bath at 5 °C for 1 min and 40 s [28].

### 2.5. Sperm Quality Assessment

For the evaluation of semen quality, a set of parameters was measured in both fresh and different thawed semen treatments.

#### 2.5.1. Macroscopic Sperm Evaluation

A precision balance was used to measure the volume through the ejaculate mass. A photometer (AccuRead, IMV Technology, L’Aigle, France) was used to measure the absorbance of each sample, which allows the calculation of the sperm concentration.

#### 2.5.2. Sperm Motility

Sperm motility and kinematic parameters were measured using a Computer Assisted Sperm Analyser (CASA) IVOS 12.3 (Hamilton Thorne Bioscience, Beverly, MA, USA). Sperm cells were diluted in the diluent (Fraction A) to 25 × 10^6^ spz/mL (final concentration) and after 5 min incubation at 5 °C, 5 µL samples were placed in a fixed-height Life Optic Chamber for sample evaluation. The motility parameters tested were: total motility (TM, %), progressive motility (PM, %), curvilinear velocity (VCL, μm/s), straight-line velocity (VSL, μm/s), average path velocity (VAP, μm/s), straightness (STR, %), linearity (LIN, %), amplitude of lateral head displacement (ALH, μm), and beat/cross frequency (BCF, Hz). Cells were considered spermatozoa when they reached an area of between 2 and 60 μm^2^ and categorized as progressive motile when VAP > 50 μm/s and STR > 70%.

#### 2.5.3. Sperm Morphology

Sperm morphology was assessed by eosin–4igrosine stain [29]. For this, a total of 0.67 g Eosin Y (Panreac, Barcelona, Spain) and 0.9 g sodium chloride (Panreac, Barcelona, Spain) were dissolved in bi-distilled water (100 mL) under gentle heating, and then 10 g 4igrosine (Panreac, Barcelona, Spain) was added. A 10 µL drop of sperm sample was mixed with a 10 µL drop of stain on a glass slide and the smear was made. For evaluation, 200 sperm cells were analyzed (×1000 magnification; Olympus, Tokyo, Japan). The percentage of spermatozoa with abnormalities in the head, mid, and tail was quantified.

#### 2.5.4. Membrane Functionality

Hypo-osmotic swelling test (HOST) was used to determine the functional membrane status of spermatozoa [30]. From each sperm fraction, 25 µL were diluted into 500 µL of hypo-osmotic solution (1 g sodium citrate, and 100 mL bi-distilled water; 100 mOsmol/kg) and warmed at 37 °C for 30 min. After incubation, samples were fixed in 2% glutaraldehyde and observed under phase contrast microscopy (×400 magnification). The sperm membrane was considered intact and functional when the sperm tail exhibited coiling. A total of 200 sperm cells were analyzed and the results were expressed as a percentage of positive endosmosis.

#### 2.5.5. Flow Cytometry

Flow cytometry analyses were carried out with a CyFlow^®^ Cube 6 Cytometer (Sysmex Europe GmbH, Norderstedt, Germany), which is composed of a 488 nm blue laser and a 638 nm red laser and features interchangeable optical filters. This device features three fluorescence channels, such as FL1 (536/40 nm bandpass), FL2 (570/50 nm bandpass), and FL3 (675 nm lowpass), together with forward scatter detection (FSC, trigger parameter) and side scatter detection (SSC). In most cases, due to the lack of specific protocols for rooster sperm, modified protocols adapted and validated to the avian species were followed [31,32,33]. In addition, the cell population was previously defined by particle size selectivity (FSC) and spore granularity (SSC) gates without the use of a fluorochrome. A maximum of 10,000 events were analyzed in each evaluation.

##### Viability

A LIVE/DEAD viability kit (Molecular Probes Europe, Leiden, The Netherlands) was used to analyze sperm viability. For this, 200 µL of semen (20 × 10^6^ spz/mL) was diluted with 300 µL (dilution 2:5) of cytometer fluid (BD FACSFlow™, BD Biosciences, San Jose, CA, USA), and 5 µL of SYBR-14 (2 µM) and 20 μL of propidium iodide (PI, 480 µM) were added. The sample was then incubated for 15 min in the dark. Before analyzing, 1200 µL of cytometer fluid was added. Spermatozoa emitting at the green wavelength (FL1) were considered to have intact plasma membranes.

##### Acrosome Integrity

For acrosome integrity assessment, 300 µL of semen (20 × 10^6^ spz/mL) was placed in a cytometer tube, and 15 µL of fluorescein isothiocyanate-conjugated peanut agglutinin (FITC-PNA, 100 µg/mL; Sigma-Aldrich, St. Louis, MO, USA) and 30 μL of PI (6 µM) were added. The sample was incubated for 5 min in the dark and 1200 µL of cytometer fluid was added before analyzing.

After carrying out both evaluations, the results were corrected as proposed by Petrunkina and Waberski [34], as these two authors estimated that within the selected population, there are events that do not correspond to spermatozoa.

##### Lipid Peroxidation (LPO)

For the assessment of lipid peroxidation (LPO), 200 µL of semen (20 × 10^6^ spz/mL) was placed in a cytometer tube and 10 µL of C11-BODIPY^581/591^ (10 µM) was added and incubated for 30 min at 37 °C in the dark. After this time, the sample was centrifuged at 681× *g* for 5 min, the supernatant was removed, and 1000 µL of cytometer fluid was added. Sperm emitting at the green wavelength (FL1) were considered positive for C11-BODIPY^581/591^. The results for this parameter are expressed as mean fluorescence intensity (MFI).

##### Reactive Oxygen Species (ROS)

For the measurement of reactive oxygen species (ROS), a commercial kit, DCFH-DA (Sigma-Aldrich, St. Louis, MO, USA), was used. For this, 1000 µL of semen (4 × 10^6^ spz/mL) was placed in a cytometer tube and 1 µL of DCFH-DA (25 µM) was added and incubated in the dark for 30 min at 25 °C. The sample was then centrifuged at 681× *g* for 5 min, the supernatant was removed, and 1000 µL of cytometer fluid was added for reading. The results for this parameter are expressed as MFI.

##### Glutathione

For the evaluation of glutathione, 1000 µL of semen (4 × 10^6^ spz/mL) was deposited in a cytometer tube and 0.5 µL of CMFDA CellTracker™ (5 µM; Molecular Probes Europe, Leiden, The Netherlands) was added. The mixture was incubated for 30 min at 37 °C in the dark. The sample was centrifuged at 681× *g* for 5 min, the supernatant was removed, and 1000 µL of cytometer buffer was added for evaluation. The results for this parameter are expressed as MFI.

CMFDA CellTracker™ can freely pass through cell membranes. Once inside the cells, these dyes are transformed into cell-impermeant reaction products. Moreover, the dyes contain a bromomethyl or chloromethyl group that reacts with thiol groups, through a glutathione S-transferase–mediated reaction, except for CellTracker Deep Red, which reacts with protein or amine groups.

### 2.6. Statistical Analysis

#### 2.6.1. Overall Descriptive Statistic

Descriptive statistics were used to summarize data obtained in fresh sperm and the control, HT1, HT2, and HT3 cryopreservation treatments for the following variables: viability, acrosome integrity, glutathione, ROS, LPO, HOST, morphology, TM, PM, and all the kinematic parameters. The descriptive statistics routine of the data description package of XLSTAT 2022 (Pearson Edition; Addinsoft, Paris, France) was used to perform this analysis.

#### 2.6.2. Discriminant Canonical Analysis (DCA)

For the DCA analysis, the following 16 explanatory variables were included: viability, acrosome integrity, glutathione, ROS, LPO, HOST, normal forms, TM, PM, VAP, VSL, VCL, ALH, BCF, STR, and LIN. As classification criteria to measure the variability between groups, different treatment HT concentrations were used. In addition, fresh sperm was included to fit better the differences between treatments.

##### Multicollinearity Preliminary Testing

Before performing DCA, the multicollinearity assumption was tested to avoid the inclusion of redundant variables that could affect the structure of the matrices or overinflate variance explanatory potential [35,36]. The variance inflation factor (VIF) was used as a multicollinearity indicator and estimated following this formula:VIF = 1/1 − R^2^,(1)

As suggested by Rogerson [37], the recommended maximum VIF value is 5. Therefore, variables with VIF values higher than 5 should be removed from further analyses. Equally, when the tolerance (1 − R^2^) of some variables is lower than 0 and VIF value ≥ 10, multicollinearity must be considered troublesome [36]. Multicollinearity analysis was computed through the discriminant analysis routine of the data analysis package of XLSTAT 2022 (Pearson Edition).

##### Canonical Correlation Dimension Determination

For the interpretation of the canonical correlations, Pearson’s ρ was used. The maximum number of canonical correlations between two sets of variables is the number of variables in the smaller set. All canonical correlations must be considered. However, the first canonical correlation explains most of the relationships between sets. Values of ≥0.30 may be indicative of a statistically significant canonical correlation dimension.

##### DCA Efficiency

Variables that significantly contribute to the discriminant function were evaluated with Wilks’ Lambda test. The contribution of variables in the discriminant function increases as Wilks’ Lambda values are closer to 0. The statistical significance of Wilks’ Lambda was assessed by the chi-square statistic. Significances smaller than 0.05 indicated that the function adequately explains the group adscription [38].

##### Independent Factor Discriminant Potential Evaluation

A better discriminating power is indicated by greater values of F and consequently, lower values of Wilks’ Lambda. This analysis was computed as a subroutine of the discriminant analysis routine of the data analysis package of XLSTAT 2022 (Pearson Editions).

##### DCA Model Reliability

The assumption of equal covariance matrices was tested in the discriminant function analysis by Pillai’s trace criterion [39]. This is the only acceptable test to be used in cases of unequal sample sizes and was computed as a subroutine of the discriminant analysis routine of the data analysis package of XLSTAT 2022 (Pearson Editions). A significance value of ≤0.05 indicates that the set of predictors considered in the discriminant model is statistically significant; hence, the application of discriminant canonical analysis is feasible.

##### Canonical Coefficient and Loading Interpretation and Spatial Representation

The percentage of allocation of a sample within its group (defined by the cryopreservation treatment) was computed through DCA. Discriminant loading values of ≥|0.40| in a variable can be considered to be significantly discriminant. Larger values for absolute coefficients of a variable determine better discriminating power. Mahalanobis distances were computed and converted into a Euclidean distance matrix. The underweighted paired-group method arithmetic averages (UPGMA) from the Rovira i Virgili University, Tarragona, Spain, and the Phylogeny procedure of MEGA × 10.0.5 from the Institute of Molecular EvolutionaryGenetics, The Pennsylvania State University, State College, PA, USA were used to build dendrograms.

##### Discriminant Function Cross-Validation

The hit radio is defined as the percentage of correctly classified cases. The leave-one-out cross-validation was used since this procedure validates the discriminant functions. Classification accuracy is considered when the classification rate is at least 25% higher than obtained by chance. *Press Q*’ statistics are used to compare the cross-validated function discriminant power:*Press Q*′ = [*n* − (*n*′*K*)]^2^/*n*(*K* − 1),(2)
where *n* is th number of observations in the sample; *n*′ is the number of observations correctly classified; and *k* is the number of groups.

The *Press Q*′ statistic value must be compared with the critical value of 6.63 for χ^2^ with a degree of freedom at a significance of 0.01. When *Press Q*′ exceeds this limit, the cross-validated classification can be regarded as significantly better than chance.

#### 2.6.3. Data Mining CHAID DT

Since most of the differences were observed between fresh sperm samples and frozen–thawed samples and not between the different HT treatments of the frozen–thawed samples, the fresh sperm samples were excluded from the CHAID DT data mining method. This tool is used for classification, prediction, interpretation, and manipulation of data. The decision support algorithm on which this method is based includes a root node, branches, and leaf nodes. Each internal node is built around an observation trait (quality parameters analyzed), while a chi-square test significance split criterion (*p* < 0.05) is fulfilled (pre-pruning).

## 3. Results

### 3.1. Overall Descriptive Statistic

The morphology, membrane functionality, and flow-cytometry-related results for fresh and different treatments of frozen–thawed sperm samples are shown in Table 1. Table 2 shows the results for motility and kinematic parameters. Viability, acrosome integrity, HOST, TM, PM, VAP, VSL, VCL, and ALH values were lower in frozen–thawed samples than in fresh sperm samples. Nevertheless, small variations were appreciable for values of these parameters between samples with different antioxidant concentrations. Otherwise, little variation was observed in the values for glutathione, ROS, LPO, morphology, BCF, STR, and LIN variables between fresh and frozen–thawed samples.

In this subsection, descriptive statistics aim to describe the midpoint of a series of scores, often referred to as a measure of central tendency (mean) and the spread of scores known as standard deviation (SD). Descriptive statistics allow an overall assessment of the data to be made to decide which of the following statistical analyses should be performed next to get the most out of the data.

### 3.2. Discriminant Canonical Analysis (DCA)

Viability, acrosome integrity, PM, VAP, VSL, VCL, and LIN variables showed VIF values higher than 5 and were discarded from further analyses (Table 3).

Wilks’ lambda test determined that the functions can be used to explain group adscription (Table 4) and a significant Pillai’s trace criterion (Table 5) determined the validity of the discriminant canonical analysis.

In Table 6, the discriminating ability of the different variables studied is shown. Variables with a high discriminating power show a high F value and consequently, low Wilks’ Lambda values. This analysis indicated that the following variables contributed significantly (*p* < 0.05) to the discriminant functions: TM, HOST, ALH, glutathione, and ROS. Conversely, LPO, BCF, STR, and normal forms did not significantly contribute to the discriminant functions (*p* > 0.05).

Four functions were revealed after the discriminant analysis (Table 7). The discriminatory power of the F1 function was high (eigenvalue of 11.043) and explained 94.455% of the variance (Figure 1).

The substitution of the values for treatments in the first two discriminating functions was performed to obtain *x*- and *y*-axis coordinates for the first and second dimensions, respectively. Coordinates were used to depict the different treatments on a territorial map (Figure 2). Centroids represent the means of the discriminant function scores by treatments for each function calculated. In this regard, Mahalanobis distance represents the probability that an observation presenting an unknown background belongs to a particular group. It can be computed through the relative distance of the problem observation to the centroid of its closest group. Then, the hit ratio was calculated. Mahalanobis distances obtained after the evaluation of the discriminant analysis matrix were transformed into squared Euclidean distances following Hair et al. [40] (Figure 3).

The purpose of this subsection is mainly to show the results obtained in the DCA, which, in turn, seeks to determine the differential clustering patterns of the different freezing treatments used in this experiment: fresh, control, HT1, HT2, and HT3. Moreover, a ranking has been established of the variables that have the greatest discriminating power when this analysis is performed, with TM (Lambda = 0.301, F = 26,173), HOST (Lambda = 0.338, F = 22,065), and ALH (Lambda = 0.442; F = 14,180) being the variables that are placed in the first positions of this ranking.

### 3.3. Data Mining CHAID DT

The data mining CHAID DT obtained from the chi-square dissimilarity matrix is shown in Figure 4. Chi-square-based branch and node distribution suggested observations differed across treatments. The first classification depended on ROS (MFI), and eight groups were depicted (824–897; 897–982; 982–1103; 1103–1175.5; 1175.5–1228.5; 1228.5–1292.5; 1292.5–1363.5; 1363.5–1442). After this classification, the observations were sorted into subgroups according to glutathione, LPO, STR, and ALH variables.

As shown in Figure 4, the decision tree starts with a root node, which has no incoming branches. The outgoing branches from the root node feed the internal nodes (the first internal node that appears is the ROS variable), which are also known as decision nodes. Depending on the available features, both types of nodes perform evaluations to form homogeneous subsets, which are indicated by leaf nodes or terminal nodes. The leaf nodes represent all possible outcomes within the data set.

## 4. Discussion

Seminal cryopreservation is an effective method for the conservation of animal genetic resources. However, there are some limitations in avian species because of the high incidences of cryoinjuries, which can lead to lethal and sublethal damage to spermatozoa [2,41,42]. Due to the characteristics of avian spermatozoa, they are highly susceptible to oxidative stress. This leads to DNA damage and decreased motility in the spermatozoa and thus, reduced fertility [42]. The high production of ROS causes lipid peroxidation of sperm membranes [43]. To neutralize this effect, the addition of antioxidants to the extenders is helpful.

In this respect, HT is an important phenolic compound present in the fruit of the olive (*Olea europea*), which has been isolated from alperujo olive pulp. This substance consists of a simple phenol that possesses a high antioxidant property and performs functions such as reducing the oxidation of low-density lipoproteins, minimizing lactate dehydrogenase activity, and protecting against H_2_O_2_ cytotoxicity [12]. Moreover, HT has anti-inflammatory and antibacterial properties [44]. Although there are studies that have evaluated the efficacy of adding HT in other species such as rats [45], humans [46], pigs [47], rams [48], and goats [49], there is no evidence of its testing in roosters. For this reason, studies testing the effect of adding this type of antioxidant to poultry semen extenders could play a pivotal role in the development of poultry reproduction techniques.

However, the preliminary results obtained through descriptive statistics in the present study suggested that there is no improvement in viability, acrosome integrity, glutathione, ROS, LPO, HOST, morphology, motility, and kinematic parameters after the freezing–thawing process in samples with supplementation of different HT concentrations (Table 1 and Table 2). Conversely, an improvement of viability, acrosome integrity, HOST, TM, PM, VAP, VSL, VCL, and ALH variables is suggested when comparing the fresh sperm samples with the frozen–thawed samples.

These results were also observed by Arando Arbulu et al. [49], who reported that there was no improvement in goat frozen semen quality when HT and another minor simple phenol such as 3,4-dihydroxyphenylglycol (DHPG) were added to the extender for semen freezing. However, according to results obtained in other species, HT addition to rat semen extender causes an improvement in sperm viability and motility parameters in fresh epididymal sperm [45].

An increase in TM has been also reported in rams when a moderate concentration of HT was added to the extender [50]. In this same species, Arando et al. [48] observed that supplementation of semen with HT improves LPO values of frozen–thawed samples as well as in the case of DHPG and the mixture of both. Similarly, Li et al. [51] have reported that the addition of 120 μmol/L of HT during boar semen storage at 17 °C has a positive effect on semen quality.

Even though no significant quality-related effects were observed in the present study, other authors have reported an improvement in the quality of semen from roosters when different levels of olive oil were added to the semen diluent during storage [52]. In particular, these improvements were noticed in viability, acrosome integrity, and motility parameters. Also, Kacel and Iguer-Ouada [53] have suggested a positive impact on the viability and motility of rooster semen when the olive oil is supplemented orally. The inclusion of olive oil in the rooster diet has also been reported to produce a decrease in the morphological injuries of the spermatozoa tail [54]. Moreover, the addition of olive oil to the diet improves gonadal activity, reduces oxidative stress and lipid peroxidation, and promotes nitric oxide signaling, thereby preserving semen quality [55]. In the same way, Eslami, et al. [56] suggest that the enrichment of semen with another derivate of the olive such as oleic acid during cooled storage would decrease the negative effects of the lipid peroxidation in rooster seminal plasma and spermatozoa. However, further studies in frozen–thawed semen samples are needed to evaluate the effect of this type of antioxidant on rooster semen quality.

The addition to extender of melatonin [57], quercitin [43], L-carnitine [58], glutamine [59], and zinc [60], among others, has shown beneficial effects in HOST variable values obtained in frozen–thawed semen. Moreover, some of these antioxidants also had a positive effect on the motility and viability variables in frozen–thawed semen.

The use of general descriptive statistics is important to provide an overview of the problem and, on this basis, to make decisions on how to address the problem. However, the development of DCA is decisive in differentiating which variables of those studied have a greater weight when evaluating the effect of supplementation of the cryopreservation extender with HT on post-thawing semen quality. After multicollinearity analysis, some variables that could present redundancy problems were eliminated (Table 3).

The sperm viability variable was discarded due to redundancy problems. Previous authors have reported a high relationship between viability as measured by flow cytometry using SYBR-14 and PI and sperm motility using the CASA system when working with stallion sperm [61,62]. For this, measurement protocols must be followed correctly and thoroughly as sometimes, a decline in TM may be reversible and may represent to a certain extent a physiologic rather than a solely pathologic change [62]. Sperm viability can also show redundancy problems with the morphology variable. There is a relationship between both variables in goat and bovine species since spermatozoa can be observed to be smaller after cryopreservation. In the case of goats, when thawing semen samples, spermatozoa show a smaller head length, width, area, and perimeter, which could also be related to the damage suffered by the sperm membrane after the osmotic differences that occur in the cryopreservation process. This leads to a reduction in the volume of the spermatozoa by changing the permeability of the membrane and thus, the cell content in the dead spermatozoa [63,64].

The acrosome integrity parameter was also eliminated in the multicollinearity test. This could be due to redundancy problems with the HOST variable. Semen manipulation, during incubation, refrigeration, and cryopreservation processes produces deterioration and changes in the acrosome and membrane of spermatozoa [65]. These changes, therefore, could produce changes in sperm motility and sperm survivability [66]. Thus, acrosome integrity during freezing can be highly correlated with most sperm-quality-related variables. In any case, the effect that an antioxidant may have on the acrosome will depend on the tested animal species and the basic composition of the cryopreservation extender supplemented [67,68]. Therefore, further studies in this field for avian species are necessary.

Finally, most of the motility-related variables were discarded since TM can synthesize and capture the variability in all these variables. Specifically, a strong correlation of VCL and VAP parameters with TM has been observed in rooster semen [69]. Additionally, several kinematic parameters are usually strongly correlated with each other [70]. Previous studies have related kinematic parameters with sperm morphology. However, the lack of standardization in measurements of motility-related traits of spermatozoa may lead to confusion about the relationship between sperm motility and sperm morphology [70,71].

After multicollinearity analysis, of the variables that remain, TM showed the highest discriminant power (Table 6). This parameter is indicative of sperm functionality and only those spermatozoa with adequate motility are able to move up the female’s oviduct to reach the sperm storage tubules and subsequently, reach the fertilization zone [72]. Sperm motility after a freeze–thaw cycle in poultry is reduced by 30 to 60% [73]. Therefore, the use of TM, which is a rapid technique, provides a wealth of information on semen status. Moreover, the economic benefit of purchasing a CASA system is viable since this tool allows an increase in the efficiency of insemination dose production and higher fertility levels [74]. Therefore, this system is becoming necessary in the daily operation of a poultry breeding center.

HOST showed a high discriminant power. This test can be used to assess the damage in the spermatozoa plasma membrane of spermatozoa due to the loss of permeability that occurs during the cryopreservation process [75]. In this sense, the state of the plasma membrane could predict the fertilizing capacity of spermatozoa as it is considered a key element in the survival of spermatozoa in the female reproductive tract by acting as a selective barrier between the intracellular and extracellular environment [76]. However, during the cryopreservation process, spermatozoa are exposed to an increasingly hypertonic medium. Therefore, an imbalance in sperm osmotic exchange occurs. The rupture of the plasma membrane supports the HOST parameter as highly variable and therefore has a high discriminating power [77].

Finally, ALH was ranked third according to its discriminant power. This parameter corresponds to the maximum width of sperm head oscillation during movement, which facilitates sperm penetration into the zona pellucida of the mature oocyte [78]. A relationship between this parameter and the total number of piglets born has been observed in porcine species [79]. In roosters, good results were obtained for the ALH parameter when the straws were frozen at 3 cm above the level of liquid nitrogen, resulting in values that did not differ from those obtained in fresh samples [80]. Thus, this could be considered the kinematic parameter of greatest interest, since it could act as a reliable predictor of the success of the in vitro fertilization process [81].

Lastly, in order to observe the differences between the sperm-quality-related traits between the different freezing treatments used in the present study, the CHAID DT was used (Figure 4). This statistical tool showed a large discriminant power of the ROS variable between the different cryopreserved treatments. In Figure 4, it can be observed that the highest ROS values are observed in samples without the presence or with the lowest tested concentrations of HT. When ROS production is excessive, the generated mechanical damage, which is unrelated to plasma membrane peroxidation, may not be repairable by the addition of exogenous antioxidants [82].

## 5. Conclusions

The present statistical method has been validated as an effective tool for understanding the importance of some parameters to evaluate the antioxidant capacity of HT when added to the cryopreservation extender of rooster sperm. TM was found to be the most discriminating variable, so the use of the CASA system could optimize to a large extent the results of a poultry breeding center, which is very important from the point of view of the conservation of poultry genetic resources. Lastly, after performing a data mining CHAID DT excluding fresh samples, ROS was found to be the most discriminating variable. The development of studies to improve techniques for the cryopreservation of genetic material in local breeds will allow the optimization of conservation programs for these endangered breeds, especially in avian species, where scientific investigation in this field is limited.

## Figures and Tables

**Figure 1 animals-13-03079-f001:**
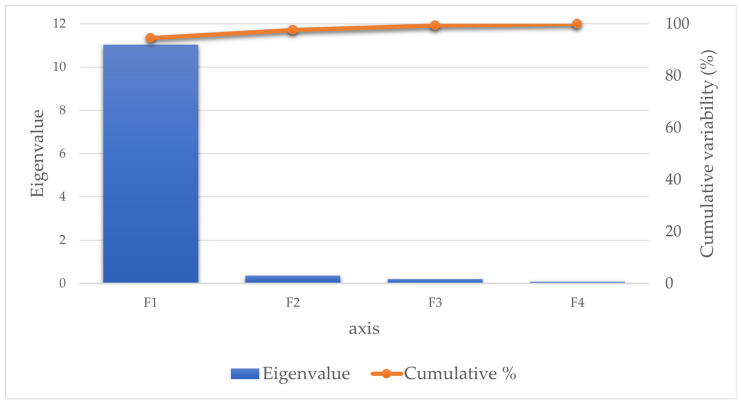
Eigenvalue and cumulative variability explanatory potential of independent explanatory variables. The eigenvalue of each discriminant function is used to measure each function’s discriminative power. The first function (F1) explains 94.46% of the variance.

**Figure 2 animals-13-03079-f002:**
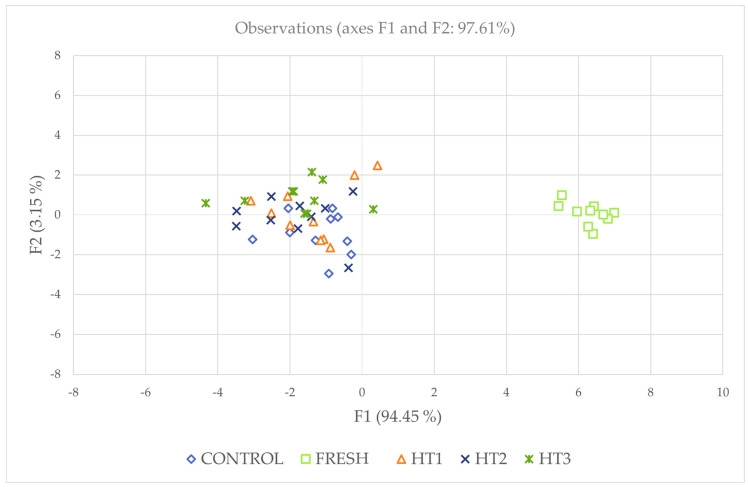
Territorial map depicting the observations belonging to each treatment considered in the DCA. The larger the differences between each point, the better the predictive power of the DCA classifying them.

**Figure 3 animals-13-03079-f003:**
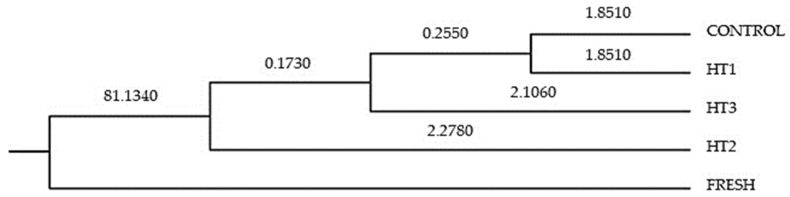
Cladogram constructed from Mahalanobis distances across different treatments. Mahalanobis distances obtained after the evaluation of the discriminant analysis matrix were transformed into squared Euclidean distances. This determines the similarity between the data belonging to each treatment.

**Figure 4 animals-13-03079-f004:**
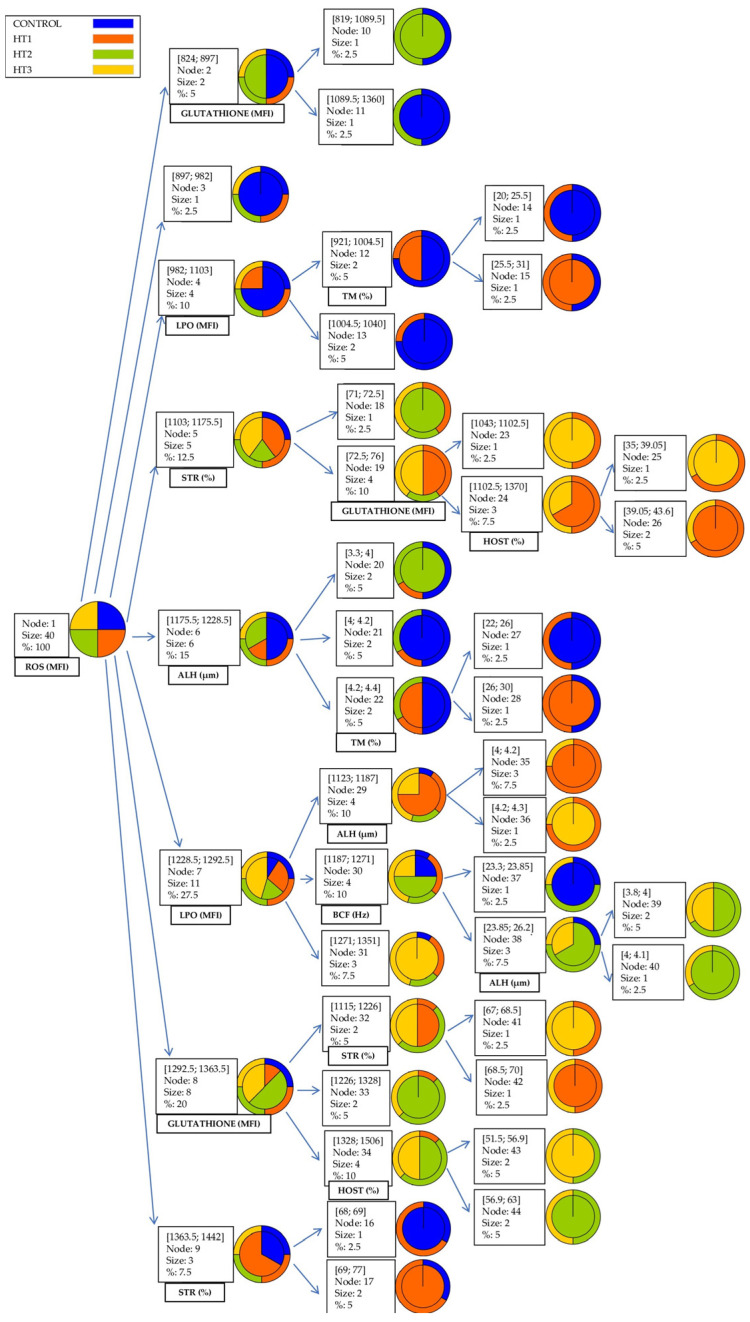
Graphic depiction of the most representative branches of the CHAID DT considering the different treatments as the clustering criterion.

**Table 1 animals-13-03079-t001:** Results of viability, acrosome integrity, glutathione, ROS, LPO, HOST, and normal forms (Mean ± SD) for the different sperm samples used.

	Fresh	Control	HT1	HT2	HT3
Viability (%)	90.33 ± 2.78	29.45 ± 12.92	28.35 ± 12.16	33.61 ± 12.96	33.01 ± 10.64
Acrosome integrity (%)	79.61 ± 8.91	22.70 ± 9.17	18.73 ± 5.65	22.76 ± 8.65	24.58 ± 9.32
Glutathione (MFI)	1090.70 ± 56.26	1197.20 ± 121.8	1257.20 ± 160.55	1238.30 ± 202.87	1303.50 ± 130.33
ROS (MFI)	1117.20 ± 34.67	1130.30 ± 171.47	1244.30 ± 113.18	1219.70 ± 158.62	1261.10 ± 63.39
LPO (MFI)	1199.80 ± 49.74	1108.70 ± 115.38	1156.30 ± 133.17	1145.30 ± 183.04	1256.40 ± 80.1
HOST (%)	79.25 ± 4.41	54.50 ± 9.19	55.01 ± 7.20	53–58 ± 8.52	53.72 ± 7.53
Normal forms (%)	76.44 ± 4.95	70.62 ± 6.67	72.15 ± 6.66	69.27 ± 8.70	69.75 ± 9.36

**Table 2 animals-13-03079-t002:** Results of TM, PM, and kinematic parameters (Mean ± SD) for the different sperm samples used.

	Fresh	Control	HT1	HT2	HT3
TM (%)	85.58 ± 2.49	42.20 ± 14.38	42.10 ± 13.50	43.40 ± 13.03	45.20 ± 11.09
PM (%)	42.86 ± 4.23	15.50 ± 6.19	16.00 ± 7.64	17.00 ± 7.26	16.10 ± 6.54
VAP (µm/s)	73.36 ± 3.27	54.46 ± 3.73	52.69 ± 4.30	54.69 ± 3.91	53.35 ± 4.86
VSL (µm/s)	57.92 ± 3.61	41.56 ± 3.07	40.75 ± 3.94	42.41 ± 3.41	40.85 ± 3.95
VCL (µm/s)	126.68 ± 3.79	99.34 ± 12.31	95.62 ± 5.88	99.91 ± 6.36	97.50 ± 9.94
ALH (µm)	4.93 ± 0.06	4.08 ± 0.56	4.09 ± 0.38	3.84 ± 0.25	4.02 ± 0.34
BCF (Hz)	25.41 ± 0.83	24.09 ± 2.57	24.99 ± 2.75	25.37 ± 3.77	24.64 ± 2.16
STR (%)	74.44 ± 1.14	73.20 ± 4.57	73.30 ± 3.13	73.40 ± 2.07	73.10 ± 3.11
LIN (%)	45.08 ± 1.37	43.30 ± 6.07	43.30 ± 3.09	44.20 ± 2.49	43.70 ± 4.50

**Table 3 animals-13-03079-t003:** Multicollinearity analysis of sperm quality-related traits.

	Tolerance (1 − R^2^)	VIF
Glutathione (MFI)	0.253	3.957
LPO (MFI)	0.272	3.671
HOST (%)	0.338	2.961
TM (%)	0.379	2.642
ALH (µm)	0.387	2.582
ROS (MFI)	0.416	2.402
BCF (Hz)	0.574	1.742
Normal forms (%)	0.616	1.622
STR (%)	0.634	1.578

Interpretation rule of thumb: variance inflation factor (VIF) = 1 (not correlated); 1 < VIF < 5 (moderately correlated); VIF ≥ 5 (highly correlated). VIFs > 5 are not presented in the table.

**Table 4 animals-13-03079-t004:** Results of Wilks’ Lambda test (Rao’s approximation).

Lambda	F (Observed Value)	F (Critical Value)	DF1	DF2	*p*-Value	Alpha
0.047	4.928	1.501	36	140	<0.0001	0.05

F, Snedecor’s F; df1, numerator degrees of freedom for the F approximation; df2, denominator degrees of freedom for the F approximation.

**Table 5 animals-13-03079-t005:** Summary of the results of Pillai’s Trace of Equality of Covariance Matrices of Canonical Discriminant Functions.

Trace	F (Observed Value)	F (Critical Value)	DF1	DF2	*p*-Value	Alpha
1.427	2.465	1.491	36	160	<0.0001	0.05

F, Snedecor’s F; df1, numerator degrees of freedom for the F approximation; df2, denominator degrees of freedom for the F approximation.

**Table 6 animals-13-03079-t006:** Results for the tests of equality of group means to test for differences in the means across sample groups once redundant variables have been removed.

	Lambda	F	DF1	DF2	*p*-Value
TM (%)	0.301	26,173.000	4.00	45.00	<0.0001
HOST (%)	0.338	22,065.000	4.00	45.00	<0.0001
ALH (µm)	0.442	14,180.000	4.00	45.00	<0.0001
Glutathione (MFI)	0.780	3176.000	4.00	45.00	0.022
ROS (MFI)	0.787	3039.000	4.00	45.00	0.027
LPO (MFI)	0.837	2188.000	4.00	45.00	0.086
Normal forms (%)	0.881	1516.000	4.00	45.00	0.214
BCF (Hz)	0.962	0.449	4.00	45.00	0.773
STR (%)	0.972	0.321	4.00	45.00	0.862

F, Snedecor’s F; df1, numerator degrees of freedom for the F approximation (groups minus 1); df2, denominator degrees of freedom for the F approximation (observations minus 1).

**Table 7 animals-13-03079-t007:** Canonical variate pairs (discriminant functions), percentage of self-explained, and cumulative variance computed in the DCA.

	F1	F2	F3	F4
Eigenvalue	11.043	0.368	0.199	0.081
Discrimination (%)	94.455	3.151	1.699	0.696
Cumulative (%)	94.455	97.606	99.304	100.000

## Data Availability

All data stemming from the present research are contained in the tables and figures. Any additional data can be obtained from the corresponding authors upon reasonable request.

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
