# Peer review of "Discriminant Analysis and Data Mining CHAID Decision Tree as Tools to Evaluate the Buffering Effect of Hydroxytyrosol on Reactive Oxygen Species in Rooster Sperm Cryopreservation"

_animals, 2023, doi:10.3390/ani13193079_

Round 1

Reviewer 1 Report

Dear authors,

attached you can find my comments and suggestions for improving your manuscript.

In my opinion, you have done nice work but especially in the section Abstract and Results extensive editing must be performed moreover with improvement of the graphical representation.

Afterward, your paper will be clear also for people from clinical practice and researchers who are not focused on statistics too much. Your work has the potential to change views on the results from this type of studies and it possibly can help to improve cryo protocols for wider spectra of the species. 

My best regards.

Reviewer 2 Report

Dear Authors,

My comments regarding the review of this paper concern mainly the methodology and description of the results:

The authors thoroughly described the procedures for preparing samples for analyzes using a flow cytometer, but this description lacks information about the analysis itself in the cytometer. How many events were counted per sample? What lasers was the cytometer equipped with? It is also worth explaining how the proper functioning of the flow cytometer was checked?

Specify what cytometer fluid (buffer) was used to dilute the samples (line 189). Was the cytometric fluid  purchased from the manufacturer or did the authors prepare it themselves? If yes, its composition should be provided.

Pearson's correlation coefficient should be entered in the statistical analysis instead of Peason's? (line 246).

In my opinion, in the analysis of the results of all variables (presented in tables 1 and 2), significant differences between the means should be marked. The description of the results shows that they occurred mainly between the means for fresh samples and the means for thawed samples. These differences should be marked in the tables (with letters or asterisks) and the test used to compare these means should be given. Interestingly, after thawing, there were no significant differences between the means of the different variants. Were there no significant differences for the variables between the variants in this analysis, or were they not just marked in the tables?

In table 2, the notation of units for the presented variables (VAP - BCF) should be checked, which should be consistent with the description of the methodology.

In the analysis of Glutathione, ROS and LPO (Tables: 1, 3 and 6) the values are given in units - MFI? In the description of the method, it should be clarified what is this unit?

In the presented diagram (figure 4), the font size should be increased (if possible) to make the drawing more legible and comprehensible.

The authors conducted a correct discussion, in which they cited many reports on the positive impact of HT on quality parameters in many animal species and referred to the results obtained in their research. It is worth noting that the authors emphasized the importance of statistical analysis using various statistical tools that allowed the selection of discriminating variables for a proper assessment of the effect of HT supplementation to the diluent.

In my opinion, I find the peer-reviewed manuscript interesting and worth publishing with minor revisions.

Reviewer 3 Report

This scientific  manuscript gives a positive contribution to this scientific subject. Perhaps is very dense and somewhat confuse  to people who are not a  statistical. I thinks that this manuscript can be accepted after a moderate revision.  Tables identification is somewhat confuse, because we don´t see see any tables with results of thawed semen, namely semen parameters. I think a table summarizing significant differences concerning significant different among fresh vs frozen semen will be helpful. In text some confusion can be viewed. Some sentences (for me ) are incomplete, because i don´t know if the treatments are different among fresh semen or among thawed semen or fresh vs frozen semen. It is a major deficiency in this manuscript,  which created confusion  and became this manuscript somewhat difficult to understand. Other situation because we have different semen concentrations  which are direct linked to what semen analysis is performed, it is important how semen are dilutes, i mean which are the substances, extenders, etc?. What are the products, extenders for semen dilutions?. It is the semen extender?. It is important to say in methods. I don´t see semen results in thawed semen?.  Say in results antioxidant effects, clearly. English sentences are not clear/suitable. It creates confusion. . In line 306, what means minor differences?. If they are not significant, i suspect that they are  equal. Isn´t it?. Clarify semen concentration in line 186.  Is it a semen dilution 1:20?.  I think that a better explanation must be done in table  concerning the presented results. Lines 391-394, are somewhat confuse.  I mean, these reported results are in fresh or frozen semen?.  Line 396, idem.  (fresh or frozen)?. idem for lines 383-389; clarify  if antioxidants has significant effects (sentences are incomplete, which because this manuscript, somewhat confuse).  Idem for line 418, . Line 475-477, is confuse. (english text, not perfect).

They are described in the previous chart. I think, that several sentences are incomplete, which creates confusion; other sentences have not a "good english". Perhaps the author needs to read with more attention.
